# Digital citizen science for ethical monitoring of youth physical activity frequency: Comparing mobile ecological prospective assessments and retrospective recall

Sheriff Tolulope Ibrahim[1], Jamin Patel[1,2], Tarun Reddy Katapally [ID][1,2,3]*

1 DEPtH Lab, School of Health Studies, Faculty of Health Sciences, Western University, London, Ontario, Canada, 2 Department of Epidemiology and Biostatistics, Schulich School of Medicine and Dentistry, Western University, London, Ontario, Canada, 3 Children's Health Research Institute, Lawson Health Research Institute, London, Ontario, Canada

* tarun.katapally@uwo.ca

## Abstract

Physical inactivity is a leading risk factor for mortality worldwide. Understanding youth patterns of moderate-to-vigorous physical activity (MVPA) is essential for addressing non-communicable diseases. Digital citizen science approaches, using citizen-owned smartphones for data collection, offer an ethical and innovative method for monitoring MVPA. This study compares the frequency of MVPA reported by youth using retrospective surveys and mobile ecological prospective momentary assessments (mEPAs) to explore the potential of digital citizen science for physical activity (PA) surveillance. Youth (N = 808) were recruited from Saskatchewan, Canada, between August and December 2018. Sixty-eight participants (ages 13–21) provided complete data on retrospective surveys (International Physical Activity Questionnaire, Simple Physical Activity Questionnaire, Global Physical Activity Questionnaire) and prospective mEPAs. Wilcoxon signed-rank tests compared retrospective and prospective MVPA frequencies, while negative binomial regression analysis examined associations between contextual factors and MVPA. Significant differences were found in the frequency of MVPA reported via retrospective surveys versus mEPAs (p < 0.000). Prospective MVPA was associated with family and friend support, having drug-free friends, part-time employment, and school distance, while retrospective MVPA frequency was associated with school and strength training. Digital citizen science, utilizing mEPAs, can provide more accurate and timely data on youth MVPA. With increasing smartphone access and digital literacy, mEPAs represent a promising method for developing effective and personalized MVPA recommendations for youth. However, these findings should be interpreted with caution, as the sample represents a small subset of youth, limiting generalizability to other youth populations.

**Data availability statement:** The data that supports the findings of this study can be found in the following link: https://figshare.com/articles/dataset/Digital_citizen_science_for_ethical_surveillance_of_physical_activity_among_youth_mobile_ecological_momentary_assessments_vs_retrospective_recall/23982912?-file=45421474.

**Funding:** This research was supported by the Saskatchewan Health Research Foundation (Establishment Grant and Patient-Oriented Research Leader Award to TRK) and the Canada Research Chairs Program (Canada Research Chair in Digital Health for Equity awarded to TRK. The funders had no role in study design, data collection and analysis, decision to publish, or preparation of the manuscript.

**Competing interests:** The authors have declared that no competing interests exist.

## Author summary

Physical inactivity is a major global health concern and a leading risk factor for death worldwide. To better understand and manage non-communicable diseases, it is important to monitor how often young people engage in moderate-to-vigorous physical activity (MVPA). This study is part of the Smart Platform, a study that combines citizen science, community based participatory research and systems science to engage participants. This study compared two methods of MVPA surveillance (i.e., retrospective surveys versus real-time mobile ecological prospective assessments – mEPAs). In 2018, 68 youth aged 13–21 in Saskatchewan, Canada, used their smartphones to report their MVPA and other health related behaviour over an eight-day study period, including weekdays and weekends. This data was then used to determine the frequency of MVPA. The study found significant differences in the frequency of MVPA reported retrospectively versus mEPAs. Additionally, variations were observed in how contextual and demographic factors were associated with physical activity in both types of assessments. This suggests the need to adapt physical activity monitoring practices to leverage digital tools and engage participants through their own devices for more comprehensive surveillance.

## Introduction

Physical activity (PA) plays a critical role in the prevention and management of non-communicable diseases (NCD) among youth [1]. However, over 80% of youth do not meet the recommended PA guidelines [2]. Evidence suggests a dose-response relationship between PA and health; the more PA that youth engage in, the greater their health benefits [3]. For instance, studies have shown that youth participation in PA is associated with improved cardiovascular fitness [4], decreased risk of type 2 diabetes, improved bone health [5], reduced body fat, and stronger muscles [3,6,7]. PA also benefits mental health by improving sleep quality, reducing symptoms of depression and anxiety, and lowering substance use [8]. Among different intensities of PA, moderate-to-vigorous PA (MVPA) has shown a stronger association with cardiovascular and metabolic benefits compared to light-intensity PA [9].

Accurate MVPA measurement is essential for monitoring adherence to PA guidelines and informing policy interventions aimed at increasing youth PA engagement [10]. However, measuring MVPA is challenging due to its complex nature, which varies by intensity, frequency, duration, and mode [11]. Traditional retrospective self-report measures, commonly used in MVPA surveillance, are prone to recall biases, social desirability effects, and difficulty in tracking daily activity patterns [12,13]. These limitations highlight the need for more accurate and ecologically valid measurement approaches, particularly for youth whose PA behaviours fluctuate depending on social and environmental contexts [14,15].

Digital citizen science offers an opportunity to enhance MVPA surveillance by engaging youth as active participants in data collection [16]. The widespread adoption of ubiquitous devices such as smartphones in the Western world [17] enables real-time MVPA monitoring through mobile ecological momentary assessments (mEMAs), which capture PA behaviours in natural settings [18]. However, mEMAs require frequent prompts throughout the day, which can be disruptive for youth balancing academic and social commitments [19]. The high response burden can lead to participant fatigue, which limits the feasibility of mEMAs in youth populations [20,21]. To address these challenges, we adapted mEMAs by shifting the reporting interval to the end of the day, a method referred to as mobile Ecological Prospective Assessments (mEPAs). Compared to retrospective measures, mEPAs offer structured end-of-day reflections that may improve recall for structured and scheduled activities, such as organized sports or social PA events, while reducing participant burden [22].

Social Cognitive Theory (SCT) provides a useful framework for evaluating PA measurement approaches by emphasizing the interaction of personal, environmental, and social factors on behaviour [23,24]. Youth PA engagement is shaped by peer interactions, parental support, and access to recreational spaces – factors that vary throughout the day [25,26]. While real-time assessments like mEMAs can capture these dynamic influences, they may not fully account for self-regulation and reflection, which are critical for sustained PA engagement [23,27]. By integrating structured opportunities for self-reflection, mEPAs reinforce critical SCT mechanisms, such as goal-setting and self-monitoring, helping youth consolidate PA experiences, recognize behaviour patterns, and adjust their future actions accordingly [27].

While several studies have compared retrospective and momentary assessments of human behaviour, including sedentary behaviour [28], PA [29], and substance use [30], in adult populations, limited research has examined how youth engage with smartphones to report MVPA using both digitally deployed retrospective surveys and mEPAs. Thus, this study aims to engage youth as citizen scientists [31] to compare MVPA frequency reported via retrospective surveys and mEPAs within the same cohort. Additionally, it accounts for sociodemographic, behavioural, and contextual factors that may influence reporting differences, allowing for a more controlled comparison between methods. Given the recall biases and self-report limitations of retrospective measures [32], we hypothesize that mEPAs, which provide structured, end-of-day reflections, will yield different MVPA estimates by encouraging self-regulation and reinforcing goal-setting – key components of SCT [23,27].

## Methods

### Study design

The study included both cross-sectional and longitudinal measures [33,34] to engage with participants in Regina, an urban centre in the Canadian prairie province of Saskatchewan. This study examined MVPA behaviours and associated factors among youth who participated in the Smart Platform [35]. The Smart Platform combines citizen science, community-based participatory research, and systems science to digitally explore behavioural phenomena, engage in knowledge translation, and deploy real-time interventions [35,36] in a given population. At the core of the Smart Platform are the citizen scientists who engage with the platform through their personally owned smartphones for population health research. Ethics approval for the Smart Platform was granted by the Research Ethics Boards of the University of Regina and Saskatchewan (REB #2017–029).

The Smart Platform (Fig 1) links a frontend smartphone application (app) deployed on citizen scientists' ubiquitous devices with a backend dashboard accessed by academic scientists. Essentially, the Smart Platform consists of two key human-computer interaction interfaces: citizen interface (smartphone app) and scientist interface (dashboard where anonymized citizen-generated big data are visualized in real-time). More importantly, the Smart Platform enables remote interaction in real-time, where the citizen scientists and/or academic scientists can initiate anonymized communication.

As part of the Smart Platform, a custom-built smartphone application (app) was downloaded by participants using both iOS and Android smartphones. The app enables our research team to engage with participants over eight consecutive days [37] during this study. Through the app, participants reported their MVPA data, behavioural, contextual, demographic,

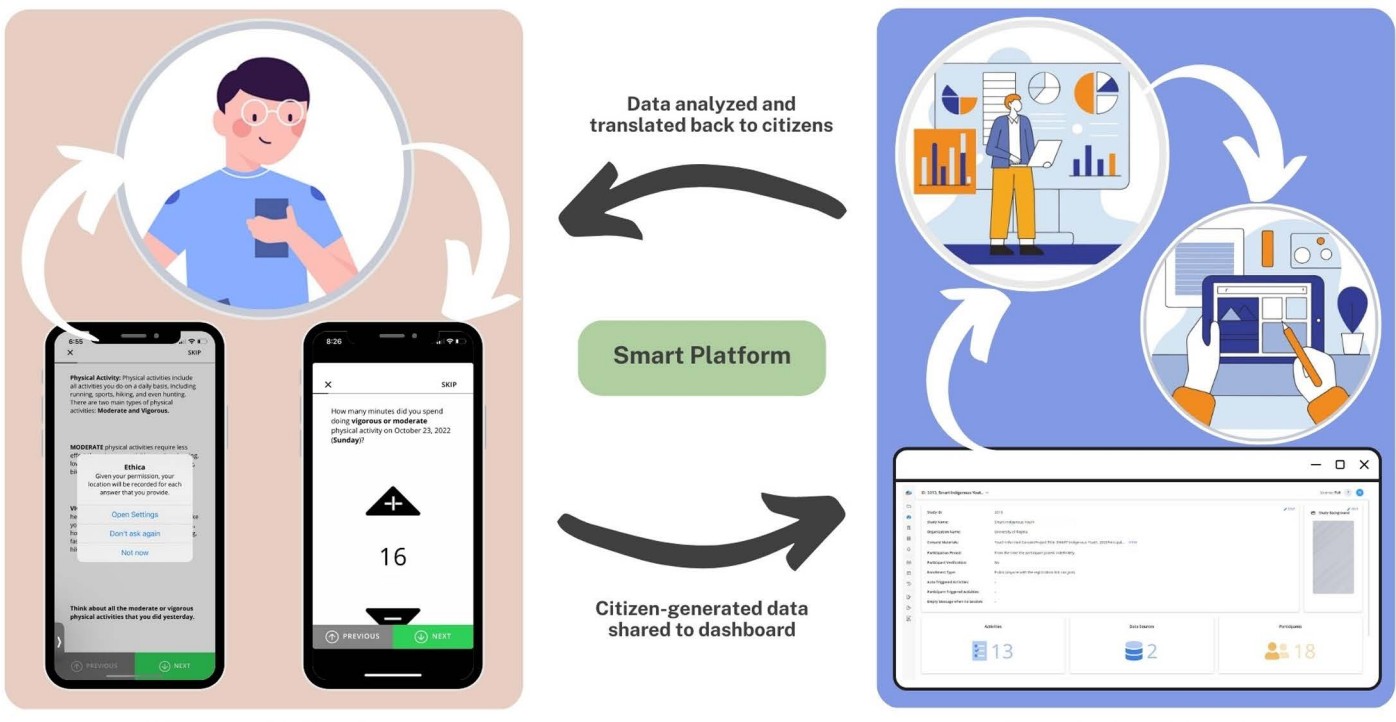

**Fig 1. The Smart Platform: citizen interface + scientist interface.**

and social factors influencing their PA behaviour [33,34,37,38]. The following data were derived from the surveys deployed through the app: peer and family support for PA, friends committed to drug-free sociodemographic characteristics, and strength training of youth citizen scientists. Before any data collection, all participants were required to confirm their age and complete the informed consent form via the app. On day 1 of joining the study, participants were required to provide responses to the modified cross-sectional validated survey measure. They were also prompted to provide responses to mEPAs from day 1 to day 8, including both weekdays and weekends, via an in-app time-triggered nudge. As this study utilized a digital citizen science approach, youth could communicate with researchers in real time via the app to provide feedback on data collection strategies, including prompt timing and the comprehensibility and clarity of survey questions [34].

## Participants

A total of 808 youth (13–21 years old) from five high schools were recruited to participate in this study between August 31st and December 31st, 2018 [39,40]. The age range of 13–21 years was chosen to capture a broad spectrum of youth during key developmental stages, from adolescence into early adulthood [41,42]. Youth were recruited through engagement sessions held in various public and catholic high schools in Regina, Saskatchewan, Canada. Using convenience sampling method, all eight high schools in the city of Regina were approached to participate in this study. Five of the eight high schools approached agreed to participate in this study. The participation rate of students in the high schools that agreed to participate was over 88% [43]. Through collaborations with school administrators, the research team scheduled in-person recruitment sessions in schools. During these recruitment sessions, the research team explained the study, demonstrated how to use the custom-built app, including how to complete mEPAs and other tasks. The research team also answered queries and concerns of youth, and assisted youth in downloading the app onto their respective smartphones.

Informed consent was provided by all youth who participated in the study through the app. Implied informed consent was obtained from parents and caregivers of participants aged 13 to 16 years old. Before in-person recruitment sessions, parents were provided the opportunity to read about the study, ask questions, and contact the research team if they did not want their child to participate. To facilitate this process, we collaborated with school administrators who emailed the informed consent forms to each household before the scheduled recruitment sessions. Parents or caregivers were required to review, sign, and provide explicit agreement for their child's participation. If we did not receive a signed consent form, we understood this as a lack of permission, indicating that the parent or caregiver did not wish for their child to participate. This approach ensured that participation was entirely voluntary, and that parental consent was obtained before involving any participant in the study.

All youth were provided with a one-week free pass to the local YMCA, valued at $25, through a partnership between our research team and the YMCA [34]. The pass was intended to provide participants with access to community-based PA resources during the study period. To avoid coercion, participants were informed that accepting the pass was optional and that their decision to participate or withdraw from the study would not affect their eligibility for the pass. Ultimately, fewer than 5% of participants used the pass, suggesting it had minimal impact on participation decisions or demographic representation [34].

## Measures

**MVPA (dependent variables).** On the first day of the study, participants were nudged through a smartphone time-triggered nudge within the app to provide retrospective MVPA data (over the previous seven days). Retrospective MVPA data were collected using a survey adapted from three validated self-reported measures: the International Physical Activity Questionnaire (IPAQ), Simple Physical Activity Questionnaire (SPAQ), and Global Physical Activity Questionnaire (GPAQ) [3,44,45]. The adaptation in the study allowed participants to report MVPA accumulation over the past seven days preceding their enrollment in the study, regardless of when they joined the study (Fig 2) [46]. For example, if a youth became a part of the study on August 31, they were prompted by the app to report their MVPA accumulation for the days of August 30, 29, 28, 27, 26, 25, and 24. Based on the number of days the participants reported engaging in MVPA in the past week, the average frequency of MVPA was calculated.

The first screen in Fig 2 defines what constitutes MVPA, while the following screens present the retrospective survey to provide citizen scientists with the opportunity to report their MVPA accumulation over the previous seven days, beginning on the first day of joining the study. Based on these responses, the frequency of MVPA for that day (will be referred to as retrospective MVPA frequency) was derived. The frequency of MVPA was defined as any day in which youth reported more than 10 minutes and less than 960 minutes (i.e., 16 hours) of MVPA following the IPAQ data processing guidelines [47].

Prospective MVPA was reported through daily time-triggered mEPAs from day 1 through day 8 of the study period including weekdays and weekends [33]. The mEPA prompts were sent once a day at a fixed time, specifically between 8:00 PM and 11:30 PM and could be completed before 12am (midnight). The timing was fixed rather than randomized to align with typical end-of-day reflection and to reduce variability in reporting windows. mEPAs had skip patterns to ensure flexibility by enabling citizen scientists to move from one question to another (Fig 3) [46].

The first screen in Fig 3 defines what constitutes MVPA, followed by a series of mEPA questions including: "What type of physical activities did you do today?" (Multiple choice) and "How many minutes did you spend doing this activity?" (Open-ended). From these questions, the frequency of MVPA for that day was derived by selecting a cut point to include engagement in at least one MVPA for 10 or more minutes (will be referred to as mEPA MVPA frequency). The dependent variables for this study included both retrospective and mEPA MVPA frequency.

**Peer support for MVPA (independent variables).** Youth were asked to consider their closest friends in the last 12 months when answering the question regarding peer support of MVPA. Peer support for MVPA was assessed with the question: "How

PLOS Digital Health

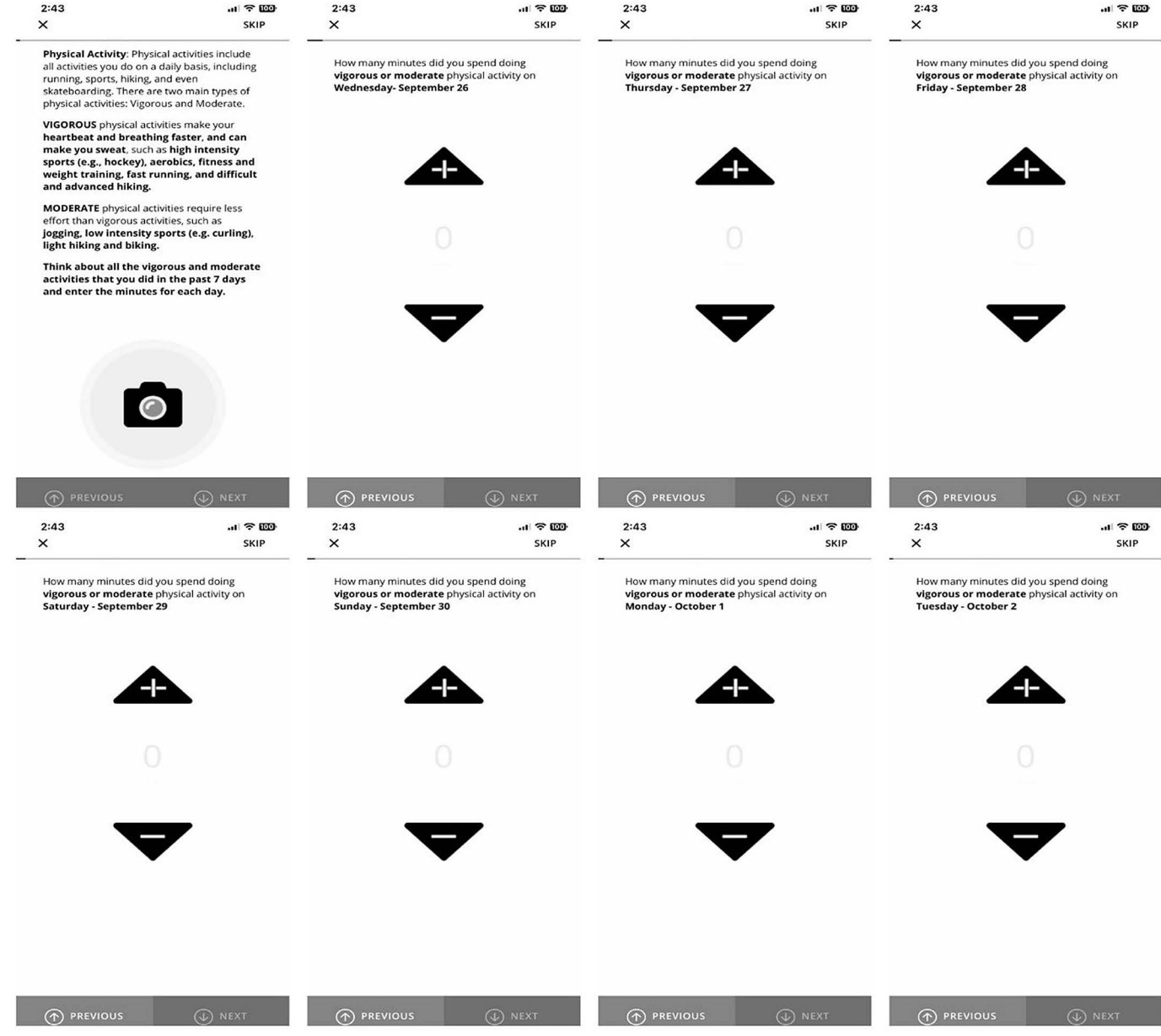

**Fig 2. Digitally deployed retrospective MVPA survey.**

many of your closest friends are physically active?" with the six response options: "none of my friends", "1", "2", "3", "4", or "5 of my friends". Responses for peer support for MVPA were dichotomized into "zero physically active friends" corresponding to "none of my friends" and "at least 1 active friend" corresponding to "1", "2", "3", "4" or "5 of my friends". This dichotomization was implemented to differentiate participants with no peer support from those with any level of peer influence.

**Family support for MVPA (independent variables).** Family support for MVPA was captured by asking Youth "How many times in a week an adult in your household encourages physical activity" with response options: "Never", "1-2 times per week", "3-4 times per week", "5-6 times per week" and "Daily". Responses were categorized into "Daily", "5-6 times per week", and "4 times or less per week" corresponding to "Never", "1-2 times per week", and "3-4 times per week".

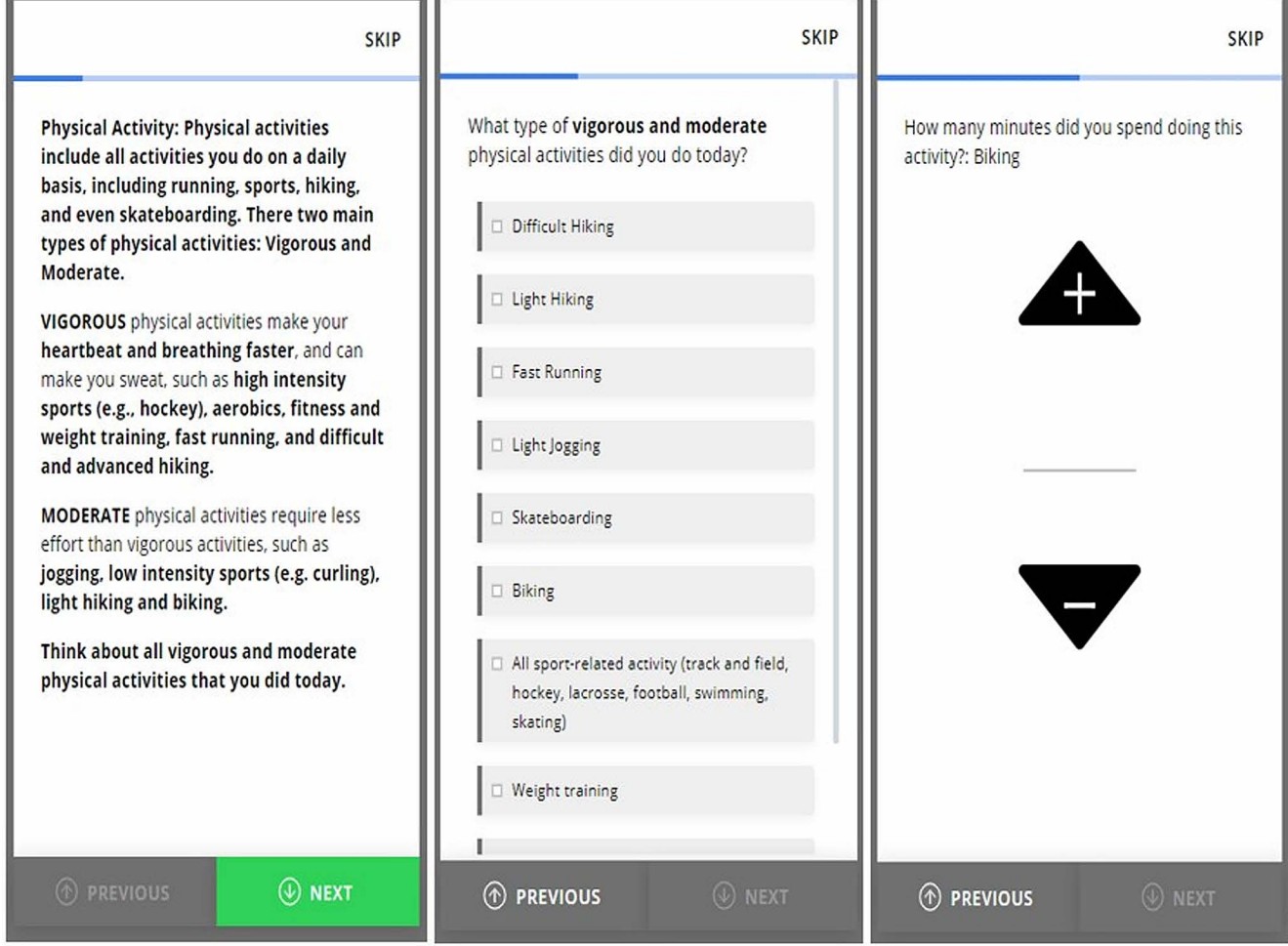

**Fig 3. Digitally deployed mEPAs.**

**Friends drug free (independent variables).**  Youth were asked to consider their friends when answering this question. Youth were asked "How many of your closest friend are committed to drug-free" with five response categories: "None of my friends", "1 of my friends", "2 of my friends", "3 of my friends" and "4 of my friends". These responses were categorized into "None of my friends", "1 of my friends", "2 of my friends" and "3 or more of my friends" which corresponds to "3 of my friends" and "4 of my friends".

**Distance from home to school (independent variables).**  Youth were asked: "To the best of your knowledge, how far is your school from home" with response categories: "Less than 2km", "Less than 5km", "Less than 10km", "Less than 20 km", and "More than 20 km". The responses were dichotomized into "Less than 5 km" which corresponds to "Less than 2km", and "Less than 5km", and "5 km or more" which corresponds to "Less than 10km", "Less than 20 km" and "More than 20 km". This dichotomization was necessary to distinguish between distances that are feasibly covered by active modes of transport (e.g., walking or cycling) and those that typically necessitate motorized transport. Also, sample distribution revealed a clustering, with a higher frequency of participants residing within 5 km of school, which supports this dichotomization

**Sociodemographic covariates.**  Gender was captured by asking participants "What is your gender?", with five response options: "male", "female", "transgender", "other (please specify)", and "prefer not to disclose". In order to

address the low counts within the categories, responses that were "transgender," "other," and "prefer not to disclose" were combined into a single category. Information on parental education was collected by asking them about the highest level of education of one of their parents or guardians through six response options: "elementary school", "some secondary/high school", "completed high school", "some post-secondary (university/college)", "received university or college degree/diploma", and "does not apply". Each of the six responses was organized into four categories of parental education: 1) "elementary school" corresponds to "elementary school or below", 2) "some secondary/high school" and "completed high school" corresponds to "at least secondary school" 3) "some post-secondary (university/college)", "received university or college degree/diploma" corresponds to "university and above" and "does not apply". The ethnicity of participants was also captured through the following response options: "First Nations", "Dene", "Cree", "Metis", "Inuit", "African", "Asian", "Canadian", "Caribbean/West Indian", "Eastern European", "European", "South Asian", "Other", and "Mixed". The responses were grouped into four categories: 1) "Indigenous" which corresponds to "First Nations", "Dene", "Cree", "Metis", "Inuit", 2) "Canadian", 3) "Mixed", and 4) "Visible minorities". The visible minorities include "African", "Asian", "Caribbean/West Indian", "Eastern European", "European", "South Asian", and "other" categories. The category "Visible minorities" was established as a result of the low number of responses within individual ethnic categories. Youth were also asked if they had a part-time job, with the following response options: "Yes" or "No". School affiliation was accounted for as a categorical variable (1–5) representing the different schools included in the study.

**Strength training (independent variables).** Engagement in strength training was measured by asking participants "On how many days in the last 7 days did you do exercises to strengthen or tone your muscles? (e.g., push-ups, sit-ups, or weight-training)". Youth were provided with eight response options including: "0 ", "1 ", "2 ", "3 ", "4 ", "5 ", "6 ", or "7 days." Each of the responses was dichotomized into "0 days of strength training" corresponding to 0 days and "at least 1 day of strength training" corresponding to "1", "2", "3", "4", "5", "6", or "7 days". This dichotomization is based on evidence suggesting that even one session per week can improve strength and help offset age-related declines [48].

**Data and risk management.** The app used in this study was designed to prioritize confidentiality, data privacy, safety, and security. All data were encrypted before being transmitted to a secure cloud server, and access to personally identifiable information on participants' smartphones (e.g., site visits, location, and contact lists) was restricted. A multi-layered security approach was implemented, including robust encryption [49] to protect against eavesdropping, man-in-the-middle attacks, and rogue hotspots [50]. To enhance control over data sharing, participants could use the "snooze" function to pause transmission. Additionally, youth data were encrypted using a hash algorithm, with MAC address anonymization, ensuring that device identifiers were irreversibly transformed to protect participant privacy. During in-person recruitment, participants were informed about privacy protections, the option to opt out, and the ability to control data uploads [34,35]. They could choose to upload data only when connected to Wi-Fi or when their devices were plugged into a power source. Clear instructions on withdrawing from the study were provided within the app, and participants could contact the research team through dedicated Smart Platform study emails for support and inquiries [34,35].

## Statistical analysis

The inclusion criteria for data to be considered for analyses was that participants had to provide MVPA data using both retrospective and prospective measures to ensure methodological rigour. Missing data points were deleted from the study analyses. A priori power analysis determined that a minimum sample size of 59 participants was required, based on an effect size of 0.15 and a power of 90%, which was lower than the number of participants who provided complete data in this study. Given the high proportion of missing MVPA data, particularly in the mEPA measures, a complete case analysis was used to minimize potential bias from imputation. While this reduced the sample size, it allowed for direct comparisons between retrospective and mEPA MVPA measures.

Data analyses for this study were conducted using R 4.2.1, an open-source statistical tool [51]. Frequencies and percentages were used to describe the categorical independent variables. Chi-square tests were used to assess the significance of difference in the proportion of independent variables included in this study. Cohen's d and Wilcoxon sign rank

test was used to ascertain the difference between retrospective and mEPA MVPA frequency. Negative binomial regression was used to assess factors associated with retrospective and mEPA frequencies. The overdispersion parameter from the Poisson regression analysis (Retrospective: 1.399; mEPA: 1.081) indicated that the data were overdispersed, making negative binomial regression a more appropriate model.

Covariates, including peer and family support [25,52], strength training [53,54], distance and friends' commitment to being drug-free, were selected based on their established associations with physical activity engagement. Peer and family support align with SCT by shaping self-efficacy and providing vicarious reinforcement, which can motivate youth to engage in MVPA [25,52]. Social factors, such as proximity to active peers and substance-free environments, have also been linked to participation in structured and recreational physical activities, aligning with SCT's emphasis on environmental determinants of behaviour. Strength training was included given its role in overall MVPA patterns and its increasing emphasis in youth exercise guidelines [53,54], though it is not explicitly part of SCT. All results were considered statistically significant at $p < 0.05$.

## Results

This study included 808 youth (aged 13–21 years); however, only 436 participants provided complete data using the retrospective MVPA measure at baseline. Moreover, after excluding participants who did not provide complete MVPA data using the prospective measure (mEPA), the final sample size was 68 participants. The summary statistics of the participants (N = 68) are summarized in Table 1. The participants were predominantly female (56.72%), with 40.30% identifying as male, and 2.98% identifying as transgender, other, or preferring not to disclose. The predominant ethnicity was Canadian (37.31%), while 35.82% identified as mixed, 25.37% identified as visible minority, and 1.50% as Indigenous. In terms of socioeconomic status, most youth (72.06%) reported that one of their parents had at least a university degree. For strength training, 81.54% of participants reported that they engaged in strength training at least once, while 18.46% reported they had never engaged in strength training. Additionally, a majority of youth reported having at least one or more physically active friends (88.88%). Further, most participants (70.15%) reported not having a part-time job. There were no significant differences between the baseline and final sample across all included variables, except for school (p = 0.008) and age (p = 0.014).

The summary statistics, Cohen's d, and Wilcoxon sign rank test of the outcome variables for this study are presented in Table 2. Youth reported an average MVPA frequency of 5.06 times per week using the retrospective measure and 2.10 times per week using the prospective (mEPA) measure. The Wilcoxon signed rank test (p < 0.001) indicated a statistically significant difference between the mean frequency of MVPA reported retrospectively and prospectively via mEPAs. Additionally, Cohen's d value (d = 1.37; 95% Confidence Interval = 0.900, 1.839) indicates a significant difference between MVPA and mEPAs.

Table 3 presents the Negative binomial regression analyses showing the association between sociodemographic and contextual factors with retrospective MVPA frequency (model 1) and prospective MVPA frequency (model 2). In the retrospective MVPA model, two contextual variables were found to be significantly associated with retrospective MVPA frequency. For instance, youth who attended school 3 (β = -1.347, [C.I.] = -2.383, -0.311, p–value = 0.010) reported a lower frequency of MVPA in comparison to youth who attended school 1. Moreover, youth who engaged in at least one day of strength training (β = 0.477, 95% [C.I.] = 0.042, 0.912, p–value = 0.031) reported a higher frequency of MVPA in comparison to youth who did not engage in strength training.

Contrastingly, in the mEPA model (i.e., prospective MVPA: model 1), several contextual variables were found to be significantly associated with the frequency of MVPA. Regarding family support for MVPA, youth whose family members encouraged PA 5–6 times per week (β = 1.294, 95% [C.I.] = 0.282, 2.306, p–value = 0.012) reported a higher MVPA frequency compared to youth whose family members encouraged PA 4 times or less per week. In contrast, youth who had family member who encouraged them to participate in MVPA daily (β = -0.699, 95% [C.I.] = -1.320, 0.078, p–value = 0.027) reported a lower frequency of MVPA in comparison to youth whose family members encouraged MVPA 4 times or less

**Table 1. Summary statistics of youth participating in this study.**

| Dependent Variables | Average frequency per week (Standard deviation) | | | | | | |
|---|---|---|---|---|---|---|---|
| Retrospective MVPA | 5.06 (2.47) | | | | | | |
| mEPA MVPA | 2.10 (1.78) | | | | | | |
| **Independent Variables (Categorical)** | **Baseline (N = 436)** | | | **Final sample (N = 68)** | | | |
| | **Frequency** | **Percentage** | **p-value** | **Frequency** | **Percentage** | **p-value** | **Attrition (%)** |
| **Gender** | | | | | | | |
| Male | 161 | 38.5 | <0.000 | 27 | 40.30 | <0.000 | 83.23 |
| Female | 233 | 55.8 | | 38 | 56.72 | | 83.69 |
| Transgender/ Other/ Prefer not to disclose | 24 | 5.7 | | 2 | 2.98 | | 91.66 |
| Total | 418 a | 100 | | 67 a | 100 | p=0.776 | 83.97 |
| **Ethnicity** | | | | | | | |
| Indigenous | 21 | 5.0 | p<0.000 | 1 | 1.50 | p<0.000 | 95.24 |
| Canadian | 166 | 39.8 | | 25 | 37.31 | | 84.94 |
| Mixed | 124 | 29.7 | | 24 | 35.82 | | 80.65 |
| Visible minority | 106 | 25.5 | | 17 | 25.37 | | 83.96 |
| Total | 417 a | 100 | | 67 a | 100 | p=0.5503 | 83.93 |
| **School***  | | | | | | | |
| 1 | 110 | 25.3 | p<0.000 | 16 | 23.5 | p<0.000 | 85.45 |
| 2 | 74 | 17.1 | | 4 | 5.90 | | 94.60 |
| 3 | 50 | 11.5 | | 4 | 5.90 | | 92 |
| 4 | 78 | 18.0 | | 2 | 32.35 | | 97.43 |
| 5 | 122 | 28.1 | | 22 | 32.35 | | 81.97 |
| Total | 434 a | 100 | | 68 | 100 | p=0.008 | 84.33 |
| **Parental education** | | | | | | | |
| At least secondary school | 91 | 21.0 | p<0.000 | 14 | 20.59 | | 84.62 |
| University and above | 282 | 65.1 | | 49 | 72.06 | | 82.62 |
| Does not apply | 48 | 11.1 | | 5 | 7.35 | | 89.58 |
| Total | 433 a | 100 | | 68 | 100 | p=0.6368 | 84.30 |
| **Distance from home to school** | | | | | | | |
| Less than 5 km | 248 | 57.54 | p=0.001 | 42 | 62.69 | p=0.038 | 83.06 |
| 5 km or more | 183 | 42.46 | | 25 | 37.31 | | 86.33 |
| Total | 431 a | 100 | | 67 a | 100 | p=0.807 | 84.45 |
| **Strength training** | | | | | | | |
| Zero days of strength training | 97 | 23.5 | p<0.000 | 12 | 18.46 | p<0.000 | 87.63 |
| At least one day of strength training | 315 | 76.5 | | 53 | 81.54 | | 83.17 |
| Total | 412 | 100 | | 65 a | 100 | p=0.429 | 84.22 |
| **Peer support for MVPA** | | | | | | | |
| Zero active friends | 47 | 11.3 | <0.000 | 8 | 12.12 | <0.000 | 82.98 |
| At least one active friend | 368 | 88.7 | | 58 | 88.88 | | 84.24 |
| Total | 415 a | 100 | | 66 a | 100 | 0.836 | 84.10 |
| **Family support for MVPA** | | | | | | | |
| 4 times or less per week | 333 | 78.35 | <0.000 | 52 | 78.79 | <0.000 | 84.38 |
| 5-6 times per week | 10 | 2.35 | | 2 | 3.03 | | 80.00 |
| Daily | 82 | 19.30 | | 12 | 18.18 | | 85.37 |
| Total | 425 a | 100 | | 66 a | 100 | 0.8348 | 84.47 |
| **Friend drug free** | | | | | | | |

*(Continued)*

**Table 1.** (Continued)

| Dependent Variables | Average frequency per week (Standard deviation) | | | | | | |
|---|---|---|---|---|---|---|---|
| None of my friends | 113 | 26.97 | <0.000 | 14 | 20.90 | <0.000 | 87.61 |
| 1 of my friends | 49 | 11.69 | | 8 | 11.94 | | 83.67 |
| 2 of my friends | 53 | 12.65 | | 8 | 11.94 | | 84.91 |
| 3 or more of my friends | 204 | 48.69 | | 37 | 55.22 | | 81.86 |
| Total | 419 [a] | 100 | | 67 | 100 | 0.7353 | 84.01 |
| **Part time job** | | | | | | | |
| Yes | 147 | 37.80 | <0.000 | 20 | 29.85 | <0.000 | 86.39 |
| No | 242 | 62.20 | | 47 | 70.15 | | 80.58 |
| Total | 389 [a] | 100 | | 67 | 100 | 0.272 | 82.78 |
| **Independent Variable (Continuous)** | **Mean** | **SD** | | **Mean** | **SD** | **p-value** | **Attrition (%)** |
| Age[*] | 16.04 | 1.80 | | 15.45 | 1.46 | 0.014 | 84.40 |

[a]Some participants did not provide a response to this question.

[*]There is a significant difference in the distribution between the baseline and final sample.

MVPA-Moderate- to-Vigorous Physical Activity; mEPA-Mobile Ecological Prospective Assessments; N-Sample size.

per week. Further, youth who had two (β = 0.870, 95% [C.I.] = 0.094, 1.645, p–value = 0.028) or three (β = 0.924, 95% [C.I.] = 0.322, 1.526, p–value = 0.003) friends who were drug-free reported higher MVPA in comparison to youth who had no friend who were drug-free. In terms of peer support, youth who had at least one active friend (β = -0.898, 95% [C.I.] = -1.633, -0.162, p–value = 0.017) reported a lower MVPA frequency in comparison to youth who had no physically active friends. Additionally, youth who did not have a part time job (β = -0.615, 95% [C.I.] = -1.173, -0.057, p–value = 0.031) reported a lower frequency of MVPA in comparison to youth who had a part time job. Youth who lived 5 km or more from their school (β = -0.615, 95% [C.I.] = -1.173, -0.057, p–value = 0.031) reported a higher frequency of MVPA compared to those who lived less than 5 km from their school.

## Discussion

This study was carried out to ascertain if there is a significant difference between the frequency of MVPA assessed through a digitally deployed retrospective survey, modified from three validated PA questionnaires, and prospectively through mEPAs. Understanding the frequency of MVPA is essential for NCD prevention and management as it has not only been associated with improved health outcomes in the youth population [55], but it is also a key component of existing PA guidelines for children and youth [56]. Further, this study explored various factors associated with the frequency of MVPA reported retrospectively and through mEPAs in the same cohort of participants (13–21 years old).

The main finding of this study was that youth reported a higher frequency of MVPA retrospectively than via mEPAs, highlighting a significant difference between these two assessment methods. This discrepancy could be attributed to recall biases inherent in retrospective PA reporting, as noted in previous research [57,58]. Recall bias has been reported to be a major contributor to the overreporting of MVPA [59], especially when the instrument is used over a longer recall period. However, not all self-reported retrospective PA instruments are prone to recall biases. For instance, some studies have pointed out that time-use diaries have the potential to reduce recall bias in PA assessments [60,61]. Additionally, self-reporting bias [62], has been reported to be a major shortcoming of retrospective self-assessed behaviour [63]. For instance, a study reported that students in grade 9–12 tend to over-report their height but under report their age [64]. Further, external bias related to social desirability has also been found to modify the reporting of self-assessed behaviour [62]. Social desirability bias is the propensity to self-report behaviour in a way that is viewed as desirable by society [65]. While social desirability bias can distort self-reported behaviour in in-person assessments [66], digitally deployed surveys may reduce this bias due to their anonymous nature [67].

Given their shorter duration of reporting periods, mEPAs can mitigate recall biases inherent in retrospective surveys [68]. In particular, collecting PA data close to the time it occurred reduces the reliance on memory recall over extended periods, thus, resulting in more reliable and accurate data [69–71]. Moreover, prospective assessments of PA, enable more frequent data collection, which provides more granular data in near-real-time [72]. Near real-time data collection minimizes the potential distortion in data reporting that may arise from retrospective recall bias [73]. Overall, the short reporting periods of mEPAs not only offer a means of minimizing recall biases but also facilitates more accurate, timely, and comprehensive data collection, thereby enhancing the reliability and validity of research findings [70]. These findings on the reported frequency of MVPA are a valuable addition to the existing literature as previous studies have not examined the differences between prospective and retrospective PA reporting among youth, particularly the frequency of MVPA.

In taking this digital citizen science approach to understanding MVPA accumulation via mEPAs, it was also important to investigate the association of MVPA frequency with sociodemographic, contextual, and behavioural factors among youth [46]. Several sociodemographic and contextual factors were considered to further understand the difference in the frequency of MVPA reported retrospectively (model 1) and via mEPAs (model 2). Our study identified five factors to be significantly associated with MVPA frequency in the mEPA model, while there were only two factors significantly associated with MVPA frequency in the retrospective model. The observed difference in the number of significant associations between the mEPA and retrospective models suggests that mEPA may be able to more effectively identify the factors associated with MVPA among youth compared to retrospective measures. However, these associations may be influenced by unmeasured seasonal and weather-related variations in activity patterns [74,75]. Fluctuations in temperature, daylight hours, and other environmental factors such as air quality are important determinants of active transportation and overall MVPA, particularly in regions with distinct seasonal changes [76–78]. Additionally, methodological advancements using cross-sectional accelerometry data have demonstrated the role of weather and seasonality on active living behaviours [75]. Future studies should integrate seasonality and objective weather measures to refine MVPA models and improve accuracy.

In the retrospective model, school enrolment was found to be significantly associated with the frequency of MVPA. However, this association was not found to be significant in prospective model. In particular, our study found that youth who attended School 3 reported a lower frequency of MVPA in comparison to youth who attended School 1. This difference could be attributable to several factors, including differences in school sports programs [79,80], physical education curricula [81], and access to PA facilities within educational institutions [26,82]. In particular, previous studies have reported that school settings such as facilities, space, equipment, and leadership support could either serve as barriers or facilitators of student healthy behaviour among students [38,83]. These study findings on the association of potential school environment with MVPA frequency reported via retrospective measures should be further explored considering the importance of school environment on youth MVPA behaviour.

Our study also found that youth who engaged in strength training reported a higher frequency of MVPA using mEPAs. Previous studies have found that individual effort in strength training was associated with retrospectively reported PA

**Table 2. Wilcoxon signed rank test showing the difference between retrospective MVPA and mEPA MVPA frequency.**

| | Average (Frequency per week) | Minimum (Frequency per week) | Maximum (Frequency per week) | N | Cohen's d 95% C. I | Wilcoxon signed rank test. (p-value) |
|---|---|---|---|---|---|---|
| **MVPA frequency** (Retrospective MVPA frequency) | 5.06 | 0 | 7 | 68 | 1.37 95% C.1 = 0.900, 1.839 | V = 1593 (p < 0.000) |
| **mEPA MVPA frequency** (Prospective MVPA frequency) | 2.10 | 0 | 7 | 68 | | |

MVPA-Moderate- to-vigorous physical activity; mEPA-Mobile Ecological Prospective assessments; C.I – Confidence interval; N-Sample size.

PLOS Digital Health

**Table 3. Negative binomial regression models showing the association between retrospective and mEPA MVPA frequency and sociodemographic and contextual factors.**

| Variable | Model 1: Retrospective MVPA Frequency[a] | Model 2: mEPA MVPA Frequency[a] |
|---|---|---|
| **School** | | |
| **1 (Ref)** | | |
| 2 | 0.317 (-0.229, 0.863) | 0.694 (-0.077, 1.465) |
| 3 | **-1.347** (-2.383, -0.311)** | -0.026 (-0.999, 0.948) |
| 4 | 0.375 (-0.112, 0.863) | -0.542 (-1.306, 0.223) |
| 5 | 0.214 (-0.253, 0.682) | -0.048 (-0.800, 0.704) |
| **Family Support for MVPA** | | |
| **4 times per week (Ref)** | | |
| 5-6 times per week | 0.586 (-0.132, 1.305) | **1.294** (0.282, 2.306)** |
| Daily | 0.178 (-0.158, 0.513) | **-0.699** (-1.320, -0.078)** |
| **Friends Drug free** | | |
| **None of my friends (Ref)** | | |
| 1 of my friends | 0.378 (-0.151, 0.907) | 0.666 (-0.116, 1.448) |
| 2 of my friends | 0.231 (-0.273, 0.735) | **0.870** (0.094, 1.645)** |
| 3 or more of my friends | 0.191 (-0.178, 0.560) | **0.924** (0.322, 1.526)** |
| **Strength Training** | | |
| **Zero days of strength training (Ref)** | | |
| At least one day of strength training | **0.477** (0.042, 0.912)** | 0.108 (-0.510, 0.726) |
| **Peer support for MVPA** | | |
| **Zero physically active friends (Ref.)** | | |
| At least one active friend | 0.402 (-0.090, 0.893) | **-0.898** (-1.633, -0.162)** |
| **Part time Job** | | |
| **Yes (Ref.)** | | |
| No | 0.061 (-0.295, 0.418) | **-0.615** (-1.173, -0.057)** |
| **Distance** | | |
| **Less than 5 km (Ref)** | | |
| 5 km or more | -0.053 (-0.349, 0.243) | **0.486** (0.001, 0.971)** |
| Constant | -2.133 (-5.016, 0.749) | 1.060 (-3.563, 5.683) |
| Observations | **62[b]** | **62[b]** |

***$p < 0.05$.

[a]Both models controlled for Age, Gender, Parental education, and Ethnicity.

[b]Six observations were deleted due to missing data.

MVPA-Moderate- to-vigorous physical activity; mEPA-Mobile Ecological Prospective assessments; C.I – Confidence interval.

duration among youth [84,85]. However, there is little evidence linking MVPA frequency reported via mEPAs with strength training among youth. As strength training relies on both the frequency and duration of activity [86], prospective measures (mEPAs) are more appropriate for its measurement as they are known to reduce recall bias [87,88] and ensure activities are recorded in near real-time, and real-world settings [89,90].

In contrast with the retrospective model, our study found that various family and peer support factors were associated with frequency of MVPA in the mEPA model. For instance, youth who had an adult in the household who encouraged PA regularly (5–6 times per week) were associated with a higher frequency of MVPA, in comparison to youth who had an

adult that encouraged PA 4 times or less per week. These findings align with previous studies which found that family and parent encouragement of PA was associated with more minutes of PA among youth [91–93]; however, their study focused on PA duration, and not frequency of MVPA. Consistent encouragement from adults may provide youth with the motivation and support needed to overcome barriers and maintain a regular exercise regimen [94]. On the contrary, our study interestingly found that youth reporting that an adult encouraged PA daily was associated with less frequency of MVPA. This finding may be explained by psychological reactance, where frequent encouragement is perceived as controlling rather than supportive, leading youth to resist PA as a way to assert autonomy [95]. Previous research has suggested that excessive parental involvement in PA can sometimes lead to negative motivational outcomes, particularly if the encouragement is perceived as pressure rather than genuine support [92,94,96]. Future studies could explore whether the tone, context, or delivery of PA encouragement influences youth engagement, as well as whether encouragement is more effective when it is self-directed rather than externally imposed.

In addition to family encouragement, our study found that having one or more physically active friends was associated with lower frequency of MVPA, which contrasts with previous studies conducted among youth [15]. One possible explanation for this unexpected finding is the role of social comparison. Some youth may compare their PA levels to those of their highly active peers, leading to feelings of inadequacy that reduce motivation to engage in PA themselves [97], making them feel inferior or less motivated to engage in MVPA themselves. This aligns with research on self-efficacy, which suggests that individuals who doubt their ability to match their peers' activity levels may be less likely to participate in PA. For instance, Schroeder et al. (2020) found that self-efficacy was associated with high MVPA levels and low sedentary behaviour among youth [98]. Future research should explore how social comparisons influence youth PA behaviours and whether interventions that build self-efficacy can mitigate these effects.

The mEPA model also found that distance to school and having a part time job were significantly associated with the frequency of MVPA. In particular, attending a school that was greater than five kilometres away from home was associated with a higher frequency of MVPA compared to those whose schools were within five kilometers from home. This difference could be attributable to youth who live further from school engaging in more active school transportation [99]. Previous evidence has found short distances to school to be associated with a higher likelihood of active transport. However, the higher probability of active transport was offset by the lower PA associated with shorter distance commutes [100]. Measurement limitations may have also influenced findings, as the absence of an "I do not know" option in some survey questions, such as school distance, could have led to incomplete or inaccurate responses. Refining survey design in future research can enhance data accuracy and better capture contextual factors associated with MVPA. Additionally, our study found that youth who had no part time job had a lower frequency of MVPA compared to those with a part time job. This difference could potentially be explained by work-related PA, as youth who have a part time job may engage in PA as a part of their job [101]. Moreover, these results align with previous evidence which indicates that youth with a part-time job engage in more MVPA than youth who do not have a part time job [102].

From a public health perspective, mEPAs provide a structured and participant-driven approach to PA monitoring [36]. Unlike passive methods, which continuously collect information and raise privacy concerns, mEPAs require participants to actively report their PA, giving them greater control over what and when they share [33–35]. This makes mEPAs a feasible option for large-scale PA surveillance, particularly in youth populations where ethical considerations around digital tracking are heightened [46]. If implemented at scale, mEPAs could help policymakers and educators identify when and where youth are least active, leading to more targeted interventions [83]. For instance, if reports show lower PA levels on Mondays and Fridays, schools could introduce structured movement breaks to maintain consistency. Municipalities could also use mEPAs to assess whether recreation programs align with youth activity patterns [26]. If evening PA is low, extending operating hours at recreation centres or improving park lighting could encourage participation.

The feasibility of implementing prospective digital assessments at scale may be influenced by disparities in smartphone access and internet connectivity, which could limit participation and introduce selection bias [103–105]. Future research

should examine how differences in technology access impact participation and data quality. Additionally, exploring alternative delivery methods, such as hybrid digital and low-bandwidth approaches could help improve accessibility and ensure broader participation in digital PA monitoring [105]. In addition to access challenges, despite increasing digital literacy levels overall, participants with lower digital literacy may struggle with app navigation, leading to missing or inconsistent data [105,106]. Providing user-friendly interfaces and technical support could help mitigate these issues. Sustained participant engagement is another key consideration, as compliance with mEPAs may decline over time due to notification fatigue, competing priorities, and survey burden [20,21]. Research on optimizing adherence strategies, such as gamification, adaptive prompting, or incentives, could enhance long-term engagement in digital PA monitoring.

## Strengths and limitations

This study adds novel evidence to our understanding of MVPA frequency among youth using advanced digital technologies. By leveraging ubiquitous devices (smartphones) for PA surveillance, this study illustrates ethical digital citizen science approaches, where citizen-owned devices can be used to monitor MVPA behaviour both prospectively (mEPAs) and retrospectively. While the digital epidemiological approach enabled large-scale recruitment and real-time engagement, it may have also placed a burden on participants. A significant proportion of youth (85.75%) did not provide complete data for prospective measures, reducing the final sample size and limiting the ability to conduct subgroup analyses, including more nuanced ethnic categories. Nevertheless, the baseline demographics of those with complete data did not differ significantly from the original sample, suggesting that external validity was not substantially compromised.

As a result of the smaller sample size, only a few covariates were included in the statistical models to minimize overfitting, which could contribute to residual confounding. This limitation is further compounded by the short study duration (8 days), which restricts the ability to capture longer-term MVPA patterns and potential fluctuations due to external factors such as school schedules, weather changes, or social influences [74,77,94]. Future studies could benefit from extending the data collection period or conducting a longitudinal cohort study (>1 year) to improve statistical power and allow for stronger conclusions about causality and directionality. Additionally, randomized controlled trials could be used to evaluate the effectiveness of mEPAs in promoting PA, assessing whether real-time monitoring and feedback mechanisms lead to sustained behavioural changes. Leveraging performance-based incremental incentives for completing mEPA prompts could also enhance engagement and reduce attrition rates in future MVPA studies [107].

In terms of study measures, retrospective MVPA data were collected for the seven days preceding the eight-day prospective data collection phase. The modified survey was based on three validated surveys [3,44,45]; however, both retrospective and prospective measures assume typical MVPA habits, which may limit generalizability. Another limitation is that adapting validated surveys into a digital format required modifications that could introduce measurement error due to differences in user interface and response burden [20,21]. Future research should validate these measures for digital administration and refine survey designs to include response options for uncertainty. Integrating objective data collection methods, such as accelerometers and pedometers, could further enhance MVPA measurement accuracy [74,108,109]. Additionally, as convenience sampling does not produce a random or representative sample of the general population, incorporating random or stratified sampling methods in future research could improve the robustness of the results.

## Conclusion

The main finding of this study showed that there was a significant difference between the frequency of MVPA reported retrospectively versus prospectively. Additionally, distance from school, family and peer support for MVPA, and friends drug free were found to have significant associations with prospectively reported MVPA, while only school and strength training were associated with retrospectively reported MVPA. Digital citizen science approaches can be used to ethically monitor MVPA frequency, not only retrospectively, but also prospectively using mEPAs. This study highlights the important differences in using retrospective vs. mEPAs measures, findings which are useful to develop appropriate prevention and

management strategies for NCDs. These findings should be interpreted within the context of the study's limited sample representation. Further research is needed to assess the difference between retrospective and mEPA measures across diverse youth populations and settings. Ultimately, this study not only transforms how we utilize digital devices ethically to understand human behaviour but also provides insights to develop potential behavioural interventions that can be deployed using the same digital devices.

## Acknowledgments

The authors acknowledge the entire Digital Epidemiology and Population Health Laboratory team (DEPtH) for their unwavering support, as well as the Canadian Institute of Health Research for their support to the DEPtH Lab and the Smart Platform.

## Author contributions

**Conceptualization:** Tarun Reddy Katapally.

**Data curation:** Sheriff Tolulope Ibrahim, Jamin Patel.

**Formal analysis:** Sheriff Tolulope Ibrahim, Jamin Patel.

**Funding acquisition:** Tarun Reddy Katapally.

**Investigation:** Tarun Reddy Katapally.

**Methodology:** Tarun Reddy Katapally.

**Software:** Sheriff Tolulope Ibrahim, Jamin Patel.

**Supervision:** Tarun Reddy Katapally.

**Writing – original draft:** Sheriff Tolulope Ibrahim, Jamin Patel.

**Writing – review & editing:** Sheriff Tolulope Ibrahim, Jamin Patel, Tarun Reddy Katapally.

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
