## [Decision Letter · Decision Letter 0]

13 Dec 2024

PDIG-D-24-00360Digital citizen science for ethical surveillance of physical activity among youth: mobile ecological prospective assessments vs. retrospective recallPLOS Digital HealthDear Dr. Katapally,Thank you for submitting your manuscript to PLOS Digital Health. After careful consideration, we feel that it has merit but does not fully meet PLOS Digital Health's publication criteria as it currently stands. Therefore, we invite you to submit a revised version of the manuscript that addresses the points raised during the review process.Please submit your revised manuscript within 60 days Feb 11 2025 11:59PM. If you will need more time than this to complete your revisions, please reply to this message or contact the journal office at digitalhealth@plos.org. Please include the following items when submitting your revised manuscript:* A rebuttal letter that responds to each point raised by the editor and reviewer(s). You should upload this letter as a separate file labeled 'Response to Reviewers '. This file does not need to include responses to any formatting updates and technical items listed in the 'Journal Requirements' section below.* A marked-up copy of your manuscript that highlights changes made to the original version. You should upload this as a separate file labeled 'Revised Manuscript with Track Changes '.* An unmarked version of your revised paper without tracked changes. You should upload this as a separate file labeled 'Manuscript '. If you would like to make changes to your financial disclosure, competing interests statement, or data availability statement, please make these updates within the submission form at the time of resubmission. Guidelines for resubmitting your figure files are available below the reviewer comments at the end of this letter. We look forward to receiving your revised manuscript. Kind regards, Yuan Lai, Ph.D.Academic EditorPLOS Digital Health Yuan LaiAcademic EditorPLOS Digital Health Leo Anthony CeliEditor-in-ChiefPLOS Digital Healthorcid.org/0000-0001-6712-6626 **Additional Editor Comments (if provided):****Reviewers' Comments:** Reviewer's Responses to Questions

**Comments to the Author**

1. Does this manuscript meet PLOS Digital Health’s publication criteria ? Is the manuscript technically sound, and do the data support the conclusions? The manuscript must describe methodologically and ethically rigorous research with conclusions that are appropriately drawn based on the data presented.

Reviewer #1: No

Reviewer #2: Yes

2. Has the statistical analysis been performed appropriately and rigorously?

Reviewer #1: No

Reviewer #2: Yes

3. Have the authors made all data underlying the findings in their manuscript fully available (please refer to the Data Availability Statement at the start of the manuscript PDF file)?

Reviewer #1: Yes

Reviewer #2: Yes

4. Is the manuscript presented in an intelligible fashion and written in standard English?

Reviewer #1: Yes

Reviewer #2: Yes

5. Review Comments to the Author

Reviewer #1: Major Concerns:

The participants are representative of very small subset of youths and the study cannot be generalized upon youths in general. Other notable issues with sampling/ study:

1. Only 68 people participated; and by the authors own comment- 273 participants are required for some level of statistical significance. Furthermore- these were selected via convenience sampling which is inherently non-random and not representative of the general population

2. All participants are from the same cityin Canada. In the absence of specifying which schools this was conducted over- it’s unclear whether participants are from the socio-economic or (Catholic) religious background

3. The study appears to be conducted at fixed point during the academic cycle. This date range is not provided; and there is no discussion on seasonality (e.g.- students are less likely to be active over winter vs. spring)

4. For some surveyed data- “Don’t know” isn’t provided as an option; for instance- when students may not know how far they live away from their schools

5. The study has had a very high attrition rate (>85%)- I’d like to see this discussed in greater length. Were there any systematic differences between those who completed the study and those who dropped out? This could significantly bias the results. Also- the study should call out whether participants were aware of the participation kickback ahead of time- as the $25 YMCA pass is likely to attract/ retain a demographic of students that is inherently very different

6. The ethnicities list does not include common demographic groups like Asians, Blacks, etc.

7. The actual survey conducted isn’t provided in the figshare link. Also, the author stated that the survey was adapted from three validated questionnaires, but need to provide more information about the adaptation process and how they ensured the validity and reliability. Did they conduct any pilot testing or cognitive interviews to assess the clarity and comprehensibility of the questions?

8. The authors need to provide more information on the specific timing and frequency cadence of the mEPA prompts. How many prompts were sent per day and when? Was the timing randomized or fixed?

Minor Concerns:

1. The abstract is very long e.g.- third sentence is 64 words long and should be split to make it more readable. The abstract should also be restructured such that it is more concise and engaging and has the study findings nearer to the start of the abstract. The abstract should also call out why this study is important and what the key findings are.

2. The studys goal is to compare data gathered from smartphone surveys vs. time triggered assessments. The author summary calls out in greater detail how these two are different.

3. The paper feels like it could be significantly trimmed down. E.g.- L222-228. This is not relevant to the study and should be removed.

4. Some sentences need to be checked; e.g.- “Youth were predominantly males” isn’t true if looking at the accompanying table.

5. Explain why the specific age range of 13-21 years was chosen vs. a tighter band like 18-29.

6. Clarify the rationale for providing a YMCA pass to participants.

Reviewer #2: The title: I recommend the title: “Digital Citizen Science for Ethical Monitoring of Youth Physical Activity: Comparing Mobile Ecological Prospective Assessments and Retrospective Recall.” Consider capitalizing the start letter for every word in the title.

Abstract:

Grammar.

Lines 52–53:

• Physical inactivity is the fourth leading risk factor of mortality globally, which could be corrected as follows: Physical inactivity is the fourth leading risk factor for mortality worldwide.

Lines 55–56:

• As digital citizen science approaches utilizing citizen-owned smartphones to ethically obtain MVPA big data can transform physical activity (PA) surveillance..., can be corrected to Digital citizen science approaches using citizen-owned smartphones to collect MVPA data ethically have the potential to transform physical activity (PA) surveillance.

Lines 57–58:

• ...using smartphone-deployed retrospective validated surveys (adapted from the International Physical Activity Questionnaire, the Simple Physical Activity Questionnaire, and the Global Physical Activity Questionnaire)... cam be corrected to...using smartphone-deployed, validated retrospective surveys adapted from the International Physical Activity Questionnaire (IPAQ), Simple Physical Activity Questionnaire (SPAQ), and Global Physical Activity Questionnaire (GPAQ)...

Line 62:

• A total of 68 participants (aged 13 to 21) provided complete data... could be correct to, Sixty-eight participants (aged 13–21) provided complete data... (I prefer spelling out numbers at the beginning of a sentence.)

Introduction:

• The rationale for comparing retrospective recall with mobile ecological prospective assessments (mEPAs) is only briefly mentioned. The introduction would benefit from a clearer justification for why this comparison is crucial, especially in a youth cohort. I recommend Strengthening the rationale by providing a more detailed explanation of the limitations of retrospective recall, particularly in youth populations, and the potential advantages of using MEPs.

• The transition from conventional ecological momentary assessments to the adapted mEPAs is somewhat abrupt. The decision to modify the reporting interval is explained, but there is limited discussion on the potential trade-offs of this change (e.g., reduced real-time accuracy). You can provide a more balanced discussion of the benefits and limitations of adapting momentary assessments to end-of-day reporting, including potential impacts on data quality.

• The use of Social Cognitive Theory is mentioned briefly without much detail. The connection between the theory and the research design could be elaborated to demonstrate how it informs the choice of assessment methods (retrospective vs. mEPAs). You must expand on the application of Social Cognitive Theory and its relevance to the selection of assessment tools and measurement of PA among youth.

• The challenges related to digital literacy differences, device access, and compliance with digital assessments are not adequately addressed. I think it’s good to discuss potential barriers to implementing digital citizen science approaches, such as disparities in access to smartphones or varying levels of digital literacy among participants.

• The hypothesis is not fully integrated with the theoretical framework or the limitations of previous research. Additionally, the study aim could be more concisely stated. You can link the hypothesis to the identified gaps in the literature and provide a concise and focused statement of the study’s objectives.

Line 103:

• Evidence is clear that physical activity (PA) is crucial in the prevention and management of non-communicable diseases (NCD) among youth. Can be changed to The evidence clearly indicates that physical activity (PA) is crucial for the prevention and management of non-communicable diseases (NCDs) among youth.

Line 106:

• ...evidence indicating the presence of a dose-response relationship between PA and health – the more PA that youth engage in, the greater their health benefits. This can be corrected to ...evidence suggests a dose-response relationship between PA and health; the more youth engage in PA, the greater the health benefits.

Line 110:

• Additionally, engagement in consistent PA has been found to improve sleep quality, reduce symptoms of depression and anxiety, and minimize substance use. Try this version, Engaging in regular PA has been shown to improve sleep quality, reduce symptoms of depression and anxiety, and decrease substance use.

Line 121:

• Measuring MVPA frequency enables direct comparisons with established guidelines, making it easier to assess adherence to recommended levels of PA. can be corrected to: Measuring the frequency of MVPA allows for direct comparisons with established guidelines, facilitating the assessment of adherence to recommended PA levels.

Line 125:

• Recall errors introduce inaccuracies in reported MVPA data, while the influence of social desirability, particularly apparent in subjective recall measures, can further distort the reliability of MVPA data. To. : Recall errors introduce inaccuracies in reported MVPA data, and the influence of social desirability, especially in subjective recall measures, can further compromise data reliability.

Line 133:

• Repeated prospective measures can address these deficiencies of existing methodologies, particularly when deployed via citizen-owned digital devices to minimize recall bias and enhance collection of environmental factors that influence MVPA. Can be : Repeated prospective measures can address the limitations of existing methodologies, especially when implemented through citizen-owned digital devices, reducing recall bias and improving the capture of environmental factors affecting MVPA.

Line 171:

• To address these concerns, we modified the momentary assessments to encompass a longer reporting interval (i.e., end of day), following Shiffman et al. can be corrected: To address these issues, we adapted the momentary assessments to include a longer reporting interval (i.e., end of day), following the approach of Shiffman et al.

Methodology:

1. Study Design:

o I recommend to Provide a brief discussion on the potential ethical issues related to digital surveillance and consent, especially for minors. Clarify how data privacy is maintained in the Smart Platform.

2. Participants and Recruitment:

o I recommend to : Acknowledge the limitations of using convenience sampling. Provide more details on how implied consent was obtained and any steps taken to ensure ethical compliance.

3. Measures and Data Collection:.

o I recommend mentioning the potential biases in self-reported measures and justify using end-of-day reporting for mEPAs as a strategy to mitigate compliance issues.

4. Ethical Considerations:

o I recommend to : Include a brief note on how incentives were balanced with ethical considerations to avoid coercion. Discuss any measures taken to ensure participants were aware of their rights and data usage.

5. Measurement of Independent Variables:

o Recommendation: Consider explaining the rationale for dichotomizing these variables. A sensitivity analysis could be conducted to evaluate the impact of this decision on the study outcomes.

6. Data Collection and Risk Management:

o Recommendation: Provide additional details on the encryption method used and discuss potential risks associated with data transfer, especially over public Wi-Fi.

7. Inclusion Criteria and Data Analyses:

o Recommendation: Clarify the method used for handling missing data (e.g., listwise deletion, imputation) and provide a rationale for the choice of Poisson regression, including any steps taken to check for overdispersion.

8. Covariate Selection:

o Recommendation: Provide a brief explanation of why these specific covariates were chosen and discuss any limitations related to unmeasured confounders.

Grammatical Recommendations:

1. Line 191:

o ...in Regina, an urban centre located in the Canadian prairie province of Saskatchewan. I recommend to this change ...in Regina, an urban center in the Canadian prairie province of Saskatchewan.

2. Line 199:

o Ethics approval for the Smart Platform was granted by the Research Ethics Board of the University of Regina and Saskatchewan. I recommend to this change Ethical approval for the Smart Platform was granted by the Research Ethics Boards of the University of Regina and Saskatchewan .

3. Line 212:

o Through the app, participants reported their MVPA data, behavioural, contextual, demographic and social factors influencing their PA behaviour. I recommend to this change Through the app, participants reported their MVPA data, as well as behavioral, contextual, demographic, and social factors influencing their PA behavior.

4. Line 218:

o Additionally, they were asked to provide responses to mEPAs from day 1 to day 8, including weekdays and weekends through an in-app time-triggered nudge. I recommend to this change They were also prompted to provide responses to mEPAs from day 1 to day 8, including both weekdays and weekends, via an in-app time-triggered nudge.

5. Line 230:

o By conducting a sample size calculation at a 90% confidence level with a 5% margin of error, it was determined that a sample size of 273 participants was required for the study. I recommend to this change A sample size calculation at a 90% confidence level with a 5% margin of error determined that 273 participants were required.

6. Line 238:

o During these recruitment sessions, the research team described the study, demonstrated how to use the custom-built app, which included demonstrations of completing mEPAs among other tasks. I recommend to this change During these recruitment sessions, the research team explained the study, demonstrated how to use the custom-built app, including how to complete mEPAs and other tasks.

7. Line 244:

o Implied informed consent was obtained from parents and caregivers of participants who were between 13 and 16 years old prior to the in-person scheduled recruitment sessions. I recommend to this change Implied informed consent was obtained from parents and caregivers of participants aged 13 to 16 before the in-person recruitment sessions.

8. Line 251:

o The retrospective MVPA data were collected through a survey adapted from three validated self-reported measures: the international physical activity questionnaire, the simple physical activity questionnaire, and the global physical activity questionnaire. I recommend to this change Retrospective MVPA data were collected using a survey adapted from three validated self-reported measures: the International Physical Activity Questionnaire (IPAQ), Simple Physical Activity Questionnaire (SPAQ), and Global Physical Activity Questionnaire (GPAQ).

9. Line 309:

o Peer support for MVPA was captured with the question: “How many of your closest friends are physically active?” can be changed to: Peer support for MVPA was assessed with the question: “How many of your closest friends are physically active?”

10. Line 363:

o On how many days in the last 7 days did you do exercises to strengthen or tone your muscles? To: On how many days in the past 7 days did you engage in exercises to strengthen or tone your muscles?

11. Line 385:

o Chi-square tests were used to test the significance difference in a proportion of independent variables included in this study. To : Chi-square tests were used to assess the significance of differences in the proportions of independent variables included in this study.

12. Line 391:

o Covariates including peer and family support, strength training, distance and friends' drug-free were included in this study based in previous related literatures. To: Covariates, including peer and family support, strength training, distance, and friends' commitment to being drug-free, were included based on previous related literature.

Results:

1. Sample Size and Data Completeness:

o Try to: Discuss the implications of the reduced sample size on the study's external validity. Consider addressing any potential biases introduced by excluding participants with incomplete data.

2. Summary Statistics and Descriptive Analysis:

o I suggest including a comparison of key demographics between the initial and final samples to evaluate representativeness and potential selection bias.

3. Outcome Variables and Statistical Tests:

o I suggest Reporting the effect size (e.g., Cohen's d or rank-biserial correlation) for the Wilcoxon signed-rank test to provide a better understanding of the magnitude of the difference.

4. Poisson Regression Analysis:

o So, I suggest addressing potential overdispersion by reporting the dispersion parameter and, if necessary, using a negative binomial regression model. Provide more context and interpretation for unexpected findings, such as the negative association between daily family support and prospective MVPA.

5. Presentation of Results:

o Recommendation: Simplify the tables by using more evident labels and footnotes to explain abbreviations. Include a brief discussion to address counterintuitive findings, such as why youth with active friends reported lower MVPA.

Grammatical Corrections

1. Line 396:

o This study involved the participation of 808 youth (13-21 years); however, when the cross-sectional survey was deployed at baseline, 436 participants provided complete data using the retrospective measure for capturing MVPA.

o Correction: This study included 808 youth (aged 13-21 years); however, only 436 participants provided complete data using the retrospective MVPA measure at baseline.

2. Line 401:

o Youth were predominantly males (40.30%) of youth, with 56.72% being females, and 2.98% of them reported being transgender, other, or preferring not to reveal.

o Correction: The participants were predominantly female (56.72%), with 40.30% identifying as male, and 2.98% identifying as transgender, other, or preferring not to disclose.

3. Line 416:

o Youth reported engaging in MVPA for an average of 5.06 times per week retrospectively and 2.10 times per week prospectively.

o Correction: Youth reported an average MVPA frequency of 5.06 times per week using the retrospective measure and 2.10 times per week using the prospective (mEPA) measure.

4. Line 433:

o With respect to family support for MVPA, youth with a family member who encouraged them to participate in PA 5 to 6 times per week reported a higher frequency of MVPA than youth whose family members encouraged MVPA 4 times or less per week.

o Correction: Regarding family support for MVPA, youth whose family members encouraged PA 5 to 6 times per week reported a higher MVPA frequency compared to those whose family members encouraged PA 4 times or less per week.

5. Line 448:

o Youth whose distance from school to their home was 5 km or more reported a higher frequency of MVPA in comparison to youth whose school is less than 5 km from their homes.

o Correction: Youth who lived 5 km or more from their school reported a higher frequency of MVPA compared to those who lived less than 5 km from their school.

6. Line 453:

o The Poisson regression analyses showing the association between retrospective MVPA frequency (model 1) and prospective MVPA frequency (model 2) with sociodemographic and contextual factors are presented in Table 3.

o Correction: Table 3 presents the Poisson regression analyses showing the associations between sociodemographic and contextual factors with retrospective MVPA frequency (model 1) and prospective MVPA frequency (model 2).

Discussion Section

1. Main Findings and Interpretation:

o Recommendation: Include a brief discussion on the practical implications of using mEPAs for MVPA monitoring in youth, especially in terms of informing public health guidelines. Address the limitations of both assessment methods, including potential biases and challenges related to participant compliance.

2. Comparison with Previous Studies:

o Recommendation: Provide a more detailed explanation for the unexpected findings, such as the negative association between having active friends and MVPA frequency. Consider discussing potential underlying mechanisms, such as social comparison effects or differences in the sample population.

3. Theoretical and Practical Implications:.

o Recommendation: Include concrete suggestions for policymakers, educators, and public health practitioners on how to utilize mEPAs for monitoring and promoting physical activity among youth.

4. Limitations of the Study:

o Recommendation: Clearly state the limitations related to the reduced sample size and potential selection bias. Discuss the challenges of using digital tools for data collection, especially among diverse youth populations with varying levels of access and digital skills.

5. Future Research Directions:

o Recommendation: Propose specific research questions or study designs that could address the limitations of the current study, such as randomized controlled trials to test the effectiveness of mEPAs in promoting physical activity.

Grammatical Corrections

1. Line 488:

o This study was carried out to ascertain if there is a significant difference between the frequency of MVPA assessed through a digitally deployed retrospective survey...

o Correction: This study aimed to determine whether there is a significant difference in the frequency of MVPA assessed using a digitally deployed retrospective survey...

2. Line 496:

o The main finding of the study was that there was a significant difference between the frequency of MVPA reported retrospectively and the frequency of MVPA reported via mEPAs.

o Correction: The main finding of the study was a significant difference in the frequency of MVPA reported retrospectively compared to that reported via mEPAs.

3. Line 508:

o Social desirability bias is the propensity to self-report behaviour in a way that is viewed as desirable by society.

o Correction: Social desirability bias refers to the tendency to self-report behavior in a manner perceived as socially acceptable.

4. Line 516:

o Near real-time data collection minimizes the potential distortion in data reporting that may arise from retrospective recall bias.

o Correction: Near-real-time data collection reduces the potential distortions in data reporting caused by retrospective recall bias.

Reference section:

• Inconsistent Formatting: For example, journal titles like "Progress in Cardiovascular Diseases" (Line 691) are not consistently italicized, and some references lack proper title case.

• Missing Access Dates for Online Sources: Some references that include URLs (e.g., Lines 686 and 713) lack access dates.

• Redundant Citations: There are redundant citations of the same source, particularly for studies by Katapally TR (e.g., Lines 754, 795, and 829). These could be combined or referenced more efficiently to avoid repetition.

• Citation Order: The references are not consistently ordered in the correct sequence based on their appearance in the text. For example, reference 51 (Line 830) appears before references 39 and 40 (Lines 795 and 799). Ensuring the correct order of citations is essential for coherence.

• Incomplete References: A few citations lack complete information. For instance, Line 998 cites Van Domelen DR's study but does not provide a DOI or complete publication details. This could hinder readers from locating the source.

6. PLOS authors have the option to publish the peer review history of their article (what does this mean? ). If published, this will include your full peer review and any attached files.

**Do you want your identity to be public for this peer review?** For information about this choice, including consent withdrawal, please see our Privacy Policy .

Reviewer #1: **Yes: ** Arihan Jalan

Reviewer #2: **Yes: ** Ibrahim Shady

---

## [Decision Letter · Decision Letter 1]

31 Mar 2025

Digital Citizen Science for Ethical Monitoring of Youth Physical Activity Frequency: Comparing Mobile Ecological Prospective Assessments and Retrospective Recall

PDIG-D-24-00360R1

Dear Dr Katapally,

We are pleased to inform you that your manuscript 'Digital Citizen Science for Ethical Monitoring of Youth Physical Activity Frequency: Comparing Mobile Ecological Prospective Assessments and Retrospective Recall' has been provisionally accepted for publication in PLOS Digital Health.

Best regards,

Yuan Lai, Ph.D.

Academic Editor

PLOS Digital Health

**Additional Editor Comments (if provided):**

Please address two minor comments from the Reviewer 1 before final submission.

**Reviewer Comments (if any, and for reference):**

Reviewer's Responses to Questions

**Comments to the Author**

1. If the authors have adequately addressed your comments raised in a previous round of review and you feel that this manuscript is now acceptable for publication, you may indicate that here to bypass the “Comments to the Author” section, enter your conflict of interest statement in the “Confidential to Editor” section, and submit your "Accept" recommendation.

Reviewer #1: All comments have been addressed

Reviewer #2: All comments have been addressed

2. Does this manuscript meet PLOS Digital Health’s publication criteria ? Is the manuscript technically sound, and do the data support the conclusions? The manuscript must describe methodologically and ethically rigorous research with conclusions that are appropriately drawn based on the data presented.

Reviewer #1: Yes

Reviewer #2: Yes

3. Has the statistical analysis been performed appropriately and rigorously?

Reviewer #1: Yes

Reviewer #2: Yes

4. Have the authors made all data underlying the findings in their manuscript fully available (please refer to the Data Availability Statement at the start of the manuscript PDF file)?

Reviewer #1: Yes

Reviewer #2: Yes

5. Is the manuscript presented in an intelligible fashion and written in standard English?

Reviewer #1: Yes

Reviewer #2: Yes

6. Review Comments to the Author

Reviewer #1: The authors have addressed all my major feedback items (e.g.- acknowledging limited generalizability, clarifying methods). Giving the paper a second read- I only have two minor nits. These should be very low overhead- and if addressed; I think the papers good to publish-

1. Were there cases where participants reported improbable MVPA values (e.g.- 16 hours per day)? Mention whether you discarded or trimmed such data. Even if there were no outliers, stating you validated this helps demonstrate rigor.

2. Consider adding a short table comparing age, gender, parental education, etc. compared by completion status (i.e.- attrition). This can appear in either the main text or supplementary materials and will help readers verify how final participation rates were not affected by these variables. You have called there were no major differences in text; but having this documented as a table would help show more rigor in your analysis as well as help with better transparency.

Reviewer #2: Thanks for your efforts, please keep it up

7. PLOS authors have the option to publish the peer review history of their article (what does this mean? ). If published, this will include your full peer review and any attached files.

**Do you want your identity to be public for this peer review?** For information about this choice, including consent withdrawal, please see our Privacy Policy .

Reviewer #1: **Yes: ** Arihan Jalan

Reviewer #2: None
